# Virtual Navigator Real-Time Ultrasound Fusion Imaging with Positron Emission Tomography/Computed Tomography for Preoperative Breast Cancer

**DOI:** 10.3390/medicina57121289

**Published:** 2021-11-24

**Authors:** Mio Mori, Kazunori Kubota, Tomoyuki Fujioka, Leona Katsuta, Yuka Yashima, Kyoko Nomura, Emi Yamaga, Junichi Tsuchiya, Tokuko Hosoya, Goshi Oda, Tsuyoshi Nakagawa, Iichiroh Onishi, Ukihide Tateishi

**Affiliations:** 1Department of Diagnostic Radiology, Tokyo Medical and Dental University, 1-5-45 Yushima, Bunkyo-ku, Tokyo 113-8510, Japan; m_mori_116@yahoo.co.jp (M.M.); leonah@jcom.home.ne.jp (L.K.); 11.ruby.89@gmail.com (Y.Y.); nomura.kyoko@kameda.jp (K.N.); ymgdrnm@tmd.ac.jp (E.Y.); tuwu11@gmail.com (J.T.); ttisdrnm@tmd.ac.jp (U.T.); 2Department of Radiology, Dokkyo Medical University Saitama Medical Center, 2-1-50 Minamikoshigaya, Koshigaya, Saitama 343-8555, Japan; kubotard@dokkyomed.ac.jp; 3Department of Surgery, Breast Surgery, Tokyo Medical and Dental University, 1-5-45 Yushima, Bunkyo-ku, Tokyo 113-8510, Japan; tocco_nkgw@yahoo.co.jp (T.H.); oda.srg2@tmd.ac.jp (G.O.); nakagawa.srg2@tmd.ac.jp (T.N.); 4Department of Comprehensive Pathology, Tokyo Medical and Dental University, 1-5-45 Yushima, Bunkyo-ku, Tokyo 113-8510, Japan; iichpth2@tmd.ac.jp

**Keywords:** breast cancer, preoperative marking, virtual navigator real-time ultrasound, fusion imaging, positron emission tomography

## Abstract

We used virtual navigator real-time ultrasound (US) fusion imaging with ^18^F-fluorodeoxyglucose positron emission tomography/computed tomography (^18^F-FDG PET/CT) to identify a lesion that could not be detected on the US alone in a preoperative breast cancer patient. Of the patient’s two lesions of breast cancer, the calcified lesion could not be identified by US alone. By fusing US with ^18^F-FDG PET/CT, which had been performed in advance, the location of the lesion could be estimated and marked, which benefited planning an appropriate surgery. The fusion of US and ^18^F-FDG PET/CT was a simple and noninvasive method for identifying the lesions detected by ^18^F-FDG PET/CT.

## 1. Introduction

Virtual navigator real-time ultrasound (US) fusion imaging is the process of aligning and juxtaposing images obtained by two or more imaging modalities [1,2]. Previous studies on US/positron emission tomography (PET) fusion imaging were performed at the following sites: Neck, axilla, supraclavicular region, chest wall, mediastinum, lung, liver, abdomen, pelvis, and extremities [1,3,4]. In these studies, this technique was used to localize lesions that were difficult to detect using techniques other than ^18^F-fluorodeoxyglucose positron emission tomography/computed tomography (^18^F-FDG PET/CT) [1,3,4].

We present the case of a patient with breast cancer that was localized by virtual navigator real-time US fusion imaging with ^18^F-FDG PET/CT. To the best of our knowledge, this is the first report to present the usefulness of US/PET fusion imaging for localizing breast lesions.

## 2. Case Presentation

The patient was a Japanese woman in her 50s (Figure 1). Previous screening mammography showed abnormalities, so she visited our hospital. In our mammography, we found an irregular mass at 4 o’clock and grouped amorphous calcifications having a background density at 2 o’clock in her left breast. Although the mass was identified by US, the calcifications were not. Vacuum-assisted biopsy for the mass revealed ductal carcinoma, and a breast marker (SenoMark™, Medicon, Osaka, Osaka, Japan) was inserted near the mass. Magnetic resonance imaging (MRI) showed a 6-mm oval mass with a breast marker at 4 o’clock in her left breast and a 9-mm linear enhancement at 2 o’clock in her left breast. ^18^F-FDG PET/CT performed for staging also showed two breast lesions, and the maximum standardized uptake values were 1.4 for the 4 o’clock lesion and 1.0 for the 2 o’clock lesion, which were visually higher than those of normal mammary glands.

The calcifications were also suspected to be malignant. Because the two lesions were in close vicinity of each other, the surgeons decided to skip the biopsy of the calcifications and remove them all at once by partial mastectomy. Although stereotaxic guidance for the purpose of marking was an alternative to localize the calcifications, this procedure has the drawbacks of patient posture and exposure to repeated mammography. Virtual navigator real-time US fusion imaging with ^18^F-FDG PET/CT was selected as a more noninvasive option compared with stereotaxic guidance. We imported the ^18^F-FDG PET/CT data into the US instrument (Aplio i900, Canon Medical Systems, Otawara, Tochigi, Japan). The breast marker inserted at the time of biopsy was used as the reference. The site corresponding to the accumulation of PET was apparently normal on US, but the morphology was suitable on CT, and we were convinced that it was consistent with the lesion. After placing skin markers on the surface of this lesion and the biopsied mass, a mammogram showed that these two lesions were correctly marked. Mammography of the surgical specimen did show that the two lesions had been successfully resected. Histologically, two non-consecutive invasive ductal carcinomas were found.

## 3. Discussion

US has a higher sensitivity and detection rate for early-stage cancers than mammography [5]. However, small breast cancers and breast cancers along the mammary gland can be difficult to detect with US. Breast cancer screening has already been attempted using MRI [6,7] and may eventually be attempted using PET, considering its good detection rate for primary lesions [8,9]. Virtual navigator real-time US fusion imaging is the process of aligning and juxtaposing images obtained by two or more imaging modalities, and it is usually used for fusion of US and MRI or US and PET/CT [1,2]. Many reports have been published about US/MRI fusion imaging, and the method is being established in several organs [2,10]. To date, this technique has been used to locate hepatic lesions that are difficult to recognize on conventional US or to treat such lesions via radiofrequency ablation [11,12]. Recently, US/MRI fusion imaging of the breast has been applied as an option for identifying MRI-detected lesions or presurgical planning [10,13,14,15]. However, US/MRI fusion was not selected for our patient because an additional MRI of the supine position was required. In our case, we had two other options, i.e., (1) MRI-guided biopsy and breast marker placement and (2) stereotaxic-guided guiding-wire placement. We did not choose any of these as they are invasive and costly options, but they are often chosen in institutions where the US/PET fusion technique is not available.

There are few reports on the clinical application of US/PET fusion imaging. Venkatesan et al. [3] reported a success rate of 86.1% (31/36) in biopsy procedures performed in different anatomical sites (i.e., neck, axilla, supraclavicular region, chest wall, mediastinum, lung, liver, abdomen, pelvis, and extremities). In a study by Krücker et al. [16], 40 needle insertions for percutaneous biopsy or radiofrequency ablation were guided by a multimodal fusion imaging technique involving conventional real-time US, CT, and an additional imaging modality (PET in seven cases) for liver, kidney, mediastinum, and lung lesions. Further, Fei et al. [17] reported that compared with standard template biopsy, US/Fluciclovine PET fusion technology could more effectively determine the presence of recurrent prostate cancer, using fewer cores in 21 patients with prostate cancer. A biopsy that targets the portion having high PET accumulation, i.e., metabolically active portion, could contribute to avoiding false negatives [1]. There are also case reports of observing soft tissues in patients with malignant melanomas [18] and blood vessels in patients with large-vessel vasculitis [19]. In our case, the US/PET fusion technique was used to localize the lesion. It was possible to change the two-screen or four-screen display of US, PET/CT, and CT in real time. The indicated site of US was considered to have the same mammary gland morphology as CT, but it could not be determined whether there was a breast cancer lesion. Proper localization was pathologically confirmed to be in the center of the resected specimen.

The image quality of PET has improved steadily in recent years due to technology improvements, such as time-of-flight technology, dedicated breast PET, and digital PET/CT, and it is expected that the breast lesions detected only on PET will increase [8,20,21]. We expect the US/PET fusion imaging technique to be useful for localizing and diagnosing breast lesions by US and to have the potential to contribute to the management of breast cancer. Because this study is a single case report, the next step would be to perform a study with large sample size.

## 4. Conclusions

We presented a case in which breast cancer was not detected but was correctly localized using virtual navigator real-time US fusion imaging with ^18^F-FDG PET/CT. This may be a noninvasive and easy method for identifying lesions detected by ^18^F-FDG PET/CT, and demand is expected to increase as PET technology improves.

## Figures and Tables

**Figure 1 medicina-57-01289-f001:**
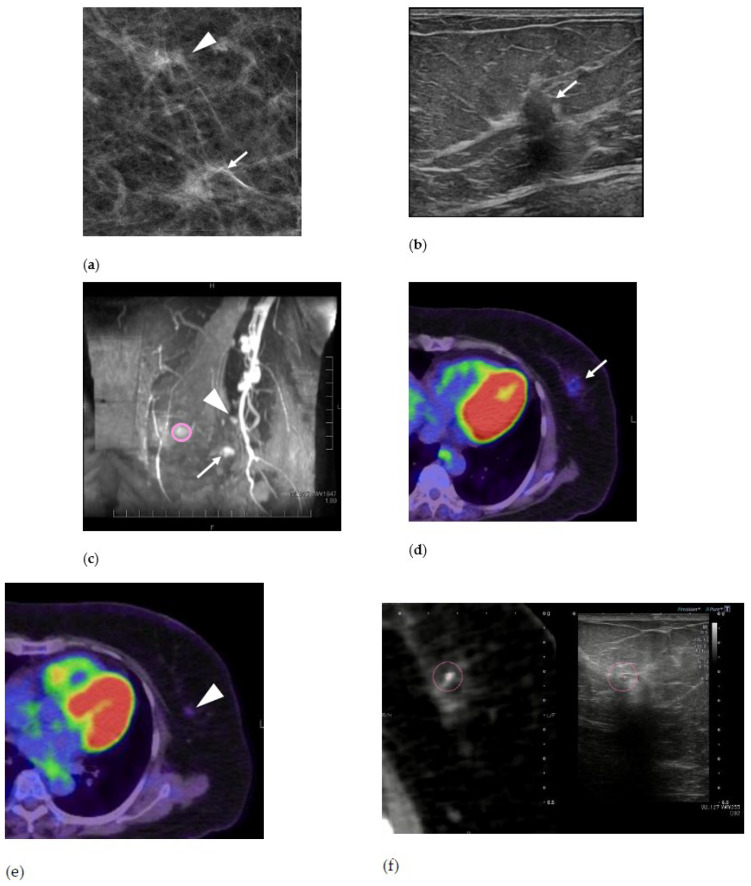
A woman in her 50s with two lesions of breast cancer in her left breast. (**a**) Mammography. Irregular mass (arrow) and grouped amorphous calcifications having a background density (arrowhead) are observed. (**b**) B-mode ultrasound (US). An irregular hypoechoic mass consistent with a mammography mass. Biopsy revealed ductal carcinoma. (**c**) Maximum intensity projection of contrast-enhanced magnetic resonance image (MRI) showing a 6-mm oval mass at 4 o’clock in her left breast (arrow) and a 9-mm linear enhancement at 2 o’clock in her left breast (arrowhead). The nipple is circled. (**d**) ^18^F-fluorodeoxyglucose positron emission tomography/computed tomography (^18^F-FDG PET/CT) fusion axial image. The mass has a breast marker nearby. The maximum standardized uptake value (SUVmax) was 1.4. (**e**) ^18^F-FDG PET/CT fusion axial image. Another 2 o’clock lesion is also found on PET/CT with an SUVmax of 1.0. (**f**) Virtual navigator real-time US fusion imaging with CT. The metallic breast marker is set as a reference (circle). (**g**) Virtual navigator real-time US fusion imaging with ^18^F-FDG PET/CT. The lesion candidate is shown on US corresponding to the accumulation of ^18^F-FDG PET/CT (circle). (**h**) Virtual navigator real-time US fusion imaging with CT. The lesion candidate shows a small amount of isolated mammary gland tissue on both US and CT (arrowheads). (**i**) Mammography for confirmation after marking. Two ring markers were placed on the skin surface (*). A metallic breast marker is shown near the irregular mass (arrow). (**j**) Mammography of the surgical specimen showing that two lesions had been resected (arrow and arrowhead). (**k**) Surgical specimen, hematoxylin and eosin staining. The lesion corresponding to the mass was a 10-mm invasive ductal carcinoma showing a cordlike and vesicular growth pattern. (**l**) Surgical specimen, hematoxylin and eosin staining. The lesion corresponding to calcification was a 6-mm invasive ductal carcinoma showing a cribriform growth pattern. Arrows, breast cancer that appeared as a mass; arrowheads, breast cancer that appeared as calcification and could be detected by MRI and ^18^F-FDG PET/CT.

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
