# Peer review of "Virtual Navigator Real-Time Ultrasound Fusion Imaging with Positron Emission Tomography/Computed Tomography for Preoperative Breast Cancer"

_medicina, 2021, doi:10.3390/medicina57121289_

Round 1

Reviewer 1 Report

It is an interesting case report, but the subject could be developed. The images are very suggestive and well captured, constituting the strength of the manuscript. 

Anyway, I suggest to insert a short introductive part before the case presentation. 

More references are needed. 

Case particularity and some conclusions are mandatory. 

Author Response

11/16/2021

Prof. Dr. Edgaras Stankevičius

Editor-in-Chief

Medicina

Thank you for inviting us to submit a revised draft of our manuscript entitled“Virtual navigator real-time ultrasound fusion imaging with positron emission tomography/computed tomography for preoperative breast cancer”to Medicina. We also appreciate the time and effort you and the reviewers has dedicated to providing insightful feedback on ways to strengthen our paper. Thus, it is with great pleasure that we resubmit our article for further consideration. We have incorporated changes that reflect the detailed suggestions you have graciously provided. We also hope that our edits and the responses we provide below satisfactorily address all the issues and concerns you and the reviewer have noted.

To facilitate your review of our revisions, the following is a point-by-point response to the questions and comments delivered in your letter dated 11/9/2021.

Reviewer 1:

Anyway, I suggest to insert a short introductive part before the case presentation.

More references are needed.

Case particularity and some conclusions are mandatory.

→Thank you for your making these suggestions. We have improved the introduction and discussion by citing further references. Following your advice, we have added some thoughts about our patients to the discussion. We would appreciate it if you could read to the manuscript because it has changed significantly.

Reviewer 2 Report

Dear authors, I have reviewed your manuscript entitled: ‘Virtual navigator real-time ultrasound fusion imaging with positron emission tomography/computed tomography for preoperative breast cancer’ in my opinion it is well-organized manuscript with substantial contents in this regard but I have one comment/suggestion:

  1. Authors should improve the structure of the article because it is very chaotic. Authors should introduce subsections such as 1) Introduction, 2) Case presentation and 3) Discussion with limitation and strength of the study 4) Conclusion in the article structure.
  2. Authors should add short introduction to the paper.
  3. Authors should discuss their observations in separate section.

Author Response

11/16/2021

Prof. Dr. Edgaras Stankevičius

Editor-in-Chief

Medicina

Thank you for inviting us to submit a revised draft of our manuscript entitled“Virtual navigator real-time ultrasound fusion imaging with positron emission tomography/computed tomography for preoperative breast cancer”to Medicina. We also appreciate the time and effort you and the reviewers has dedicated to providing insightful feedback on ways to strengthen our paper. Thus, it is with great pleasure that we resubmit our article for further consideration. We have incorporated changes that reflect the detailed suggestions you have graciously provided. We also hope that our edits and the responses we provide below satisfactorily address all the issues and concerns you and the reviewer have noted.

To facilitate your review of our revisions, the following is a point-by-point response to the questions and comments delivered in your letter dated 11/9/2021.

Reviewer 2:

Authors should improve the structure of the article because it is very chaotic. Authors should introduce subsections such as 1) Introduction, 2) Case presentation and 3) Discussion with limitation and strength of the study 4) Conclusion in the article structure.

→We agree with your opinion and have improved the structure of the article. The content of the introduction and discussion has been significantly changed for deeper consideration.

Authors should add short introduction to the paper.

Authors should discuss their observations in separate section.

→Thank you for your professional insight. We have improved the introduction and discussion by citing further references. Following your advice, we have added some thoughts about our patients to the discussion. We would appreciate it if you could read to the manuscript because it has changed significantly.

Round 2

Reviewer 1 Report

The authors significantly improved the content. 

Reviewer 2 Report

In my opinion the article has been substantially improved. Thus, I endorse publication of this article in current form.